# Dental Emergencies in an Italian Pediatric Hospital during the COVID-19 Pandemic

**DOI:** 10.3390/healthcare10030537

**Published:** 2022-03-15

**Authors:** Daniela Carmagnola, Marilisa Toma, Dolaji Henin, Mariachiara Perrotta, Laura Gianolio, Alessandra Colombo, Claudia Dellavia

**Affiliations:** 1Department of Biomedical, Surgical and Dental Sciences, Università degli Studi di Milano, Via Mangiagalli 31, 20133 Milano, Italy; daniela.carmagnola@unimi.it (D.C.); dolaji.henin@unimi.it (D.H.); mariachiara.perrotta1997@gmail.com (M.P.); claudia.dellavia@unimi.it (C.D.); 2Department of Pediatrics, Vittore Buzzi Children’s Hospital, Università degli Studi di Milano, Via Lodovico Castelvetro 32, 20154 Milano, Italy; laura.gianolio@unimi.it (L.G.); alessandra.colombo2@unimi.it (A.C.)

**Keywords:** dental emergencies, COVID-19, emergency room, oral diseases prevention, oral diseases treatment, oral health promotion

## Abstract

Emergency rooms (ER) are largely used by patients with oral complaints, who choose the ER over private or public dental offices for oral prevention and treatment. During the COVID-19 pandemic, the activity of most dental facilities was limited, and most hospitals and ERs were dedicated to the treatment of COVID-19 patients. The present study analyzed the number of and reason for visits at the emergency room (ER) of Ospedale dei Bambini “Vittore Buzzi”, the main pediatric hospital in Milano, Italy, between 2019 and 2020, with a particular focus on oral emergencies. In 2019, 25,435 children turned to the ER, compared to 16,750 in 2020. About 10% of the children were eventually admitted to the hospital in both years. The number of admissions for infectious diseases, other than COVID-19, signed an important decrease in 2020, while trauma/injuries decreased slightly in number but increased in proportion. The number and proportion of ER visits for oral complaints decreased in 2020 compared to 2019. Stomatitis was the most frequent condition, followed by traumatic injuries, which decreased in number and percentage between 2019 and 2020. Oral infections and painful caries accounted for about 15% of the cases in both 2019 and 2020. These data highlight the need to promote territorial services for the prevention and treatment of oral health complaints, including dental emergencies.

## 1. Introduction

The novel coronavirus SARS-CoV-2 was identified as the causative agent for a fast-spreading series of atypical respiratory symptoms in the Hubei Province of Wuhan, China in December of 2019. The resulting disease, named coronavirus disease 2019 (COVID-19) quickly spread to the rest of the world. Between February and April 2020, Italy experienced one of the largest clusters and case-fatality rates of COVID-19 worldwide. COVID-19 was officially declared a pandemic by the World Health Organization (WHO) on 11 March 2020 [1].

During the first Italian wave, personal movements were limited, and basically, any kind of business and activity was closed. Medical care was restricted to COVID-19-related severe symptoms or emergencies. Throughout the following spring and summer, the prevalence of COVID-19 started to decline, and most restrictive measures were progressively lifted. In late October 2020, due to a new increase in SARS-CoV-2 transmission, stay-at-home measures were issued again and gradually lifted in the late spring of 2021. Similar patterns, both concerning COVID-19 transmission trends as well as the introduction of non-pharmaceutical interventions (NPI) occurred in different countries worldwide [2,3].

During the COVID-19 pandemic and especially the first wave, most public and private territorial health care facilities in most countries were closed. Hospital departments and intensive care units prioritized COVID-19 patients, and most health care professionals were deployed to COVID-19 units. Consequently, diagnosis and treatment of other pathologies and conditions, even acute ones, were often delayed. A significantly lower number of hospital admissions of patients with myocardial infarction was reported during the lockdown compared to before the pandemic [4]. The same trend was observed for stroke [5], mental health care [6], oral cancer diagnosis [7], oncologic management [8] and diabetes [9].

Dental procedures are classified as high risk because of the oral health staff’s vicinity to the patients’ mouth and the concomitant production of aerosols [10]. Dental public services and private practices were therefore closed and/or their offer was considerably reduced during the various phases of the COVID-19 pandemics [11], either on an auto-regulatory basis (such as in Italy) or due to government restrictions, depending on the different countries’ policies [12,13].

Despite that with time, guidelines and consensuses were published concerning how to safely perform dentistry and reduce the risk for SARS-CoV-2 transmission in the dental setting [14], issues regarding the consequences of suspending routine dental care or delaying the management of oral infections and swellings have risen since the start of the pandemic [15].

Dentistry in Italy is mainly private, and children are eligible, in public facilities and free of charge, for selected basic treatments between 0 and 14 years of age. Nevertheless, the demand for public dental care overrates its supply, and preventive programs are not structured and available to all children. Consequently, many children, especially those belonging to fragile families, still suffer from advanced forms of caries and infections that might lead their parents to turn to emergency services, overloading the system for conditions that could be prevented and treated elsewhere [16,17].

To test the hypothesis that the COVID-19 pandemic might have affected some families’ decisions to turn to the ER for their kids’ dental emergencies, we analyzed the number of and reason for visits at the emergency room (ER) of Ospedale dei Bambini “Vittore Buzzi”, the main pediatric hospital in Milano, Italy, between 2019 and 2020, with a particular focus on oral emergencies. The hospital does not include a dental service.

## 2. Materials and Methods

We retrospectively analyzed data from the ER register of Ospedale dei Bambini “Vittore Buzzi”, ASST Fatebenefratelli-Sacco, Milano, Italy, during the period between 1 January 2019 and 31 December 2020. The register recorded all ER admissions, and for each of them, included date (day and time), national providence number, birthdate, gender, reason for visiting, triage emergency code (based on an adaptation of the ICD diagnosis code system: Manuale Formativo di Triage pediatrico—SIMEUP 2009; Linee di indirizzo nazionali sul triage intraospedaliero, Osservazione Breve Intensiva—OBI, sviluppo del piano di gestione del sovraffollamento in pronto soccorso. Conferenza Stato-Regioni. Agosto 2019) [18], and final diagnosis at discharge/admission. No information concerning ethnicity or any kind of socio-economic indicator were collected or could be obtained due to privacy regulations.

The files were searched by initial diagnosis name (the results of the patient’s complaint and triage) and discharge/admission final code and were singularly checked in case of errors. The following keywords were used: oral, oro-, facial, mouth, buccal, bucco-, dental, tooth, teeth, gingiva, gingival, stomatitis, aphthae, aphthous, abscess, caries, decay, lip, tongue, and cheek.

Starting from March 2019, the emergency service of the hospital introduced a pre-assessment procedure for all check-ins, excluding life-threatening conditions, in a dedicated pre-triage area, to assign children to a COVID-19 (red) or a non-COVID-19 route (blue) basing on history, signs and symptoms. In this paper, we analyzed the data concerning the blue route cases.

The numbers of overall visits and visits for oral complaints were calculated monthly, and the numbers were compared between 2019 and 2020. The visits for oral health concerns were further analyzed to allocate them to the following categories: stomatitis/aphthous stomatitis, abscesses/infections, painful caries (including any kind of pain from initial caries symptoms to pulpitis), and trauma.

The study was approved by the ethical committee of the Università degli Studi di Milano on 14 December 2021, project number 131/21.

## 3. Results

In 2019, 25,435 children turned to the ER, compared to 16,750 in 2020 (2020/2019 proportion: 65.8%). While in January and February 2020, the visits trend was similar to that of the same months of the previous year (Appendix A), and since March 2020, a clear drop was observed. About 10% of the children were eventually admitted to the hospital in both years (2058 patients, i.e., 9.8% in 2019, vs. 1818 patients, i.e., 10.8% in 2020). In 2019 and 2020, 4976 (19.6%) and 2350 (14%) patients respectively abandoned the ER before being diagnosed. The proportion of total abandoned in 2020 over 2019 was 47.2%, while admissions in 2020 were 72.5% of 2019.

Concerning the reasons for visiting, they were grouped into major categories that are displayed in Appendix A. In both years, the main reason for turning to the ER was ear pain and/or upper respiratory infections, although with different proportions (30.7% in 2019 vs. 19.4% in 2020). Trauma was the second reason (with similar numbers, namely 3452 in 2019 vs. 3007 in 2020, but more were represented in 2020 compared to 2019, namely 18.0% vs. 13.6%). These two conditions represented about 37% to 44% of all reasons for visiting in 2019 and 2020, respectively. All acute infectious diseases considered together, i.e., respiratory, ear, nose and throat, gastrointestinal and genitourinary, represented 66.5% of the reasons for visiting in 2019 compared to only 53.5% in 2020.

Focusing on oral health complaints as a reason for visiting the ER, the analysis showed that in 2019, 1.7% of the children (433 out of 25,435) turned to the ER because of oral health-related complains, compared to 1.2% in 2020 (197 out of 16,750). The monthly trends for 2019 and 2020 are shown in Appendix A. The most consistent reduction was observed during the second wave, that is, the period from October to December 2020.

In particular, in both years, stomatitis was the most frequent condition, followed by peri-oral and/or dental trauma. Infections and painful caries accounted for about 15% of the cases in both 2019 and 2020. Appendix A presents a closer analysis of the picture. In the “other” category, teething symptoms, stuck foreign bodies (mainly fishbones), eruption cysts and a vast range of similar events were included.

Concerning the therapy of infections and painful caries, all children were discharged with infection management through antibiotics and pain management through analgesics, and/or were advised to turn to a dentist for treatment.

## 4. Discussion

In the present retrospective, descriptive study, we analyzed data concerning the use of a pediatric ER before and during the COVID-19 pandemic in Italy, focusing on dental emergencies.

The overall number of visits was lower in 2020, particularly in the months characterized by COVID-19 waves and related restrictions, probably due to the lockdown period that forced people home and the fear of becoming infected. Considering the reasons for visiting the ER, the number of admissions for infectious diseases, other than COVID-19, signified an important decrease (in number and proportion) in 2020, while trauma/injuries decreased slightly in number but increased in proportion. The number and proportion of ER visits for oral complaints decreased in 2020 compared to 2019. Stomatitis was the most frequent condition, followed by traumatic injuries, which decreased in number and percentage between 2019 and 2020. Oral infections and painful caries accounted for about 15% of the cases in both 2019 and 2020.

A decline in the number of pediatric, adult and senior patients visiting the ER during 2020 and 2021, in particular during COVID-19 waves and NPI periods, together with an increase in hospital admission rates, has been reported in many countries [19,20,21,22].

Among the explanations for the decline in pediatric ER visits, some studies have reported a reduced transmission of infections due to isolation and use of personal protective devices, a trend to limit ER use unless strictly necessary, and the fear of becoming infected with SARS-CoV-2 in the ER. In our study, we calculated and compared the number of patients who abandoned the ER before being visited in 2019 and 2020. In 2020, only 14% of the patients, compared to 19.6% in 2019, left the hospital. It has been extensively reported that people often turn to the ER for minor and/or not urgent health complaints, as the ER is usually free of charge and represents an immediate and convenient alternative to local public and private services [23]. The reduction in patients leaving before being visited observed in our study might indicate that less people turned to the ER for minor and/or deferrable issues.

In our dataset, overall acute infectious diseases considered together, i.e., respiratory, ear, nose and throat, gastrointestinal and genitourinary infections, represented 66.5% of the reasons for visiting in 2019 compared to 53.5% in 2020. A reduction in ER visits due to acute infections is consistent with data from the literature [24]. A study from the Netherlands [25] reported that the largest reduction in general pediatric care in 2020 was observed for infections. Some authors, observing the reduced incidence of many viral and bacterial infections in 2020 due to isolation and delay in traditional vaccination programs, have risen the issue of an immunity debt or gap in children as a consequence of the COVID-19 lockdowns [26]. The year 2021 has seen a tremendous increase in respiratory infectious diseases caused mainly by respiratory syncytial virus [27]. Conversely, respiratory infections other than COVID-19 that share common features with SARS-CoV-2 infection might have been addressed in the red route.

When focusing on dental emergencies, we observed that both number and proportion of ER visits for oral complaints decreased in 2020 compared to 2019, despite that dental care activities were stopped or extremely limited. Stomatitis was the most frequent condition, while oral infections and painful caries accounted for about 15% of the cases in both 2019 and 2020.

Ospedale dei Bambini “Vittore Buzzi” is the main pediatric hospital in Milano, and its ER is used by several young patients also seeking care for oral complaints every year, despite that it does not include a dental unit, and therefore it cannot provide resolutive therapies for dental infections and painful caries but rather medication prescriptions to manage symptoms or infections. In our analysis, stomatitis and aphthous stomatitis accounted for more than one-third of the oral issues group in 2019, compared to about 15% in 2020. We included stomatitis and aphthous stomatitis in the oral complaints group when a more specific diagnosis did not follow: such oral lesions are an unspecific manifestation of several diseases, mainly mild and self-limiting, whose differential diagnosis can be challenging [28]. When a specific diagnosis was made for the oral manifestation, for example hand-foot-and-mouth disease, the case was included in the systemic group. Such differentiation might represent a limitation in our analysis, but at the same time, potential diagnostic mingling could not be avoided, as our ER is not a specific dental emergency department. Given that the same bias was applied to the analysis in 2019 and 2020, the decrease in the proportion of visits for stomatitis and aphthous stomatitis in 2020 may corroborate the hypothesis that many families, out of fear, choose to avoid the ER for conditions that were perceived as deferrable, non-threatening and/or self-limiting. Such explanation might apply as well to the “other” category, as unspecific symptoms related, for example to teething, might have prompted parents to turn to the ER before but not during the pandemic. Oral manifestations of COVID-19 have been described in the literature, as the oral cavity has been proven to be a susceptible anatomical site to SARS-CoV-2 infection [29]. According to Xu et al. (2020), ACE2, the receptor used by the virus to enter cells, is highly expressed in the oral mucosa [30]. SARS-CoV-2 infection might dysregulate the oral keratinocytes’ function, leading to painful oral ulcers [31]. Taste alterations and xerostomia might share similar pathogenetic features [32]. Other oral manifestations of SARS-CoV-2 infection include the exacerbation of autoimmune diseases as well as herpes simplex virus (HSV) and varicella zoster virus (VZV) infections, aphthous stomatitis, cheilitis, and tongue lesions. However, it is still not clear whether the oral symptoms are manifestations of the disease or occur due to generic physical debilitation [33]. In our sample population, we tended to exclude that the oral manifestations we observed might be linked to COVID-19, as the triage procedure excluded most positive patients. Moreover, we tended to exclude COVID-19 vaccine-related oral manifestations, as the vaccine program in Italy for kids only started in November 2021, and further studies have shown that such manifestations occur extremely rarely [34]. A study comparing COVID-19-related oral manifestations in hospitalized vs. non-hospitalized subjects has shown that such manifestations are often to be ascribed to medical devices and treatments, prone position, and immunological impairment, rather than to COVID-19 itself [35].

Ospedale Buzzi provided a telemedicine service for monitoring COVID-19 patients during the epidemic. Unfortunately, it did not provide any teledentistry services. Teledentistry has been advocated as a useful tool for oral health education and promotion, remote diagnosis and monitoring, and in pre-appointment behavior guidance, and it has proven to be particularly functional during COVID-19 [36]. Teledentisry to manage dental emergencies could help triage patients without involving unnecessary direct contact, [37], provide real-time consultation between doctor and patient [38] and, based on patient’s history and clinical symptoms, help decide whether a visit at the dental office is necessary or not [39]. In situations where dental treatment does not have to be undertaken immediately, appropriate detailed instructions for home care, including medication prescriptions, could be provided by means of teleinformation [40].

In many countries, Italy included, dental public services tend to offer only basic dental care to well-defined categories of citizens, raising availability, appropriateness, affordability, and equity issues, and turning the ER into a substitute for dental facilities. In general, patients presenting to the ER with dental complaints are estimated to account for 0.3% to over 4% of ER visits. Dental pain, dental infections, dental and maxillofacial trauma, and post-dental treatment-related complications such as hemorrhage and dry socket seem to be the most frequently reported conditions [41]. Dental attendances at medical emergency departments have been reported to be repeat visits by the same low-income patients who rely on the ER for those dental needs that could have substantial costs [42,43].

A shortcoming to visiting a general ER for dental concerns is that dental professionals might not be available, as in our case. This might contribute to explain why, whether the most appropriate or not, ER therapy for dental concerns has been reported to be often palliative and rely on the prescription of medications [44]. In a study on a large population analyzing medication prescriptions following dental-related ER visits in the United States of America, the authors reported that about 55% of the patients filled a prescription for an antibiotic and about 40% for an opioid within 7 days [45]. Another retrospective study analyzing Medicaid claims data reported that 28% of emergency dental visits ended up with no procedure, while the majority received diagnostic procedures such as radiographs or non-definitive treatments that required further appointments, often unmet, thus leading to new emergency visits, both in the adult and the pediatric population [46]. Further, it has been shown that medical staff not trained in dentistry are less precise in diagnosing and managing oral issues [41,46].

Focusing on the characteristics of dental ER use by the pediatric population, data from a university pediatric dental emergency service reported that about half of the visits were related to pain due to carious lesions, and 16.3% of the patients did not necessarily need immediate attention. This is the reason why the authors recommended a more precise definition of dental emergencies in order to prevent abuse of pediatric emergency services [47]. The results from an American pediatric dental emergency facility indicated that 40% of the visits were related to caries and that the characteristics of the patients seeking hospital care for dental emergencies were young age, non-Caucasian ethnicity, Medicaid as payer, not having a regular dentist and living close to the facility [16].

In our study, oral infections and painful caries accounted for about 15% of the cases in both 2019 and 2020. Although dental caries can be prevented and treated easily at their early stages, untreated caries can have serious consequences beyond representing an avoidable economic burden for health systems. Studies have shown that untreated caries seen at their final stage in preschoolers can require treatment under general anesthesia in 8–11% of the cases [48] and hospitalization for the administration of intravenous antibiotics in 8% of the cases [49].

Failing to meet patients’ health needs or limiting accessibility to oral health services has been associated with increased emergency visits, surgical interventions and hospital admission days [50]. According to the Global Burden of Disease 2010 study, in 2010, “untreated caries in permanent teeth was the most prevalent condition worldwide, affecting 2.4 billion people, and untreated caries in deciduous teeth was the 10th-most prevalent condition, affecting 621 million children” [51]. The burden of oral diseases in terms of economic and social impact, quality of life for those affected, and their link to social and economic inequalities and inadequate funding for prevention and treatment, particularly in low-income and middle-income countries, has been extensively discussed with the aim to highlight the urgent need to address oral diseases among other non-communicable diseases as a global health priority [52]. Despite such knowledge and awareness, strategic efforts to improve oral health are still needed, especially since the number of people with untreated oral conditions rose from 2.5 billion in 1990 to 3.5 billion in 2015, with a 64% increase in disability-adjusted life years due to oral conditions throughout the world [53].

In agreement with our findings, a study on the use of the emergency department of a medical center in Taiwan before and during the COVID-19 outbreak reported a 17% decline for dental emergencies [54]. Conversely, a Chinese study that examined data from a university department of dental emergency reported that during the COVID-19 pandemic, the number of dental emergency visits increased by almost 30% [55]. Such results seem to point to the fact that dedicated emergency dental services might be useful and should be made available for the population in order to prevent patients from giving up dental treatment [56], both during drastic events such as the recent pandemic as well as routinely.

Possible problems with the consequences of the delay in diagnosis and therapy of oral conditions due to the closure of dental activities during the pandemic have been anticipated since the first wave [12]. Recently, published data suggest that dentistry could be performed safely during the pandemic, probably as infection control is a key characteristic of dental activity and has led to the routine use of personal protective equipment and the adoption of strict sterilization and disinfection procedures [13].

The present study includes some limitations. First, diagnoses were not made by oral health professionals, and some misdiagnosing concerning oral diseases might be expected. Further, no data could be gathered concerning socio-economic characteristics, attitudes toward oral health check-ups, or eating habits of the sample population. It would have been interesting to know what kind of therapy followed the diagnoses of oral complaints, but such data are not available. Nevertheless, the same shortcomings applied before and during the pandemic.

In conclusion, our data point to the fact that ERs are frequently used for dental complaints, as reported in the literature, and that powering oral health specific preventive and emergency services might help the population to obtain adequate answers to their complaints and decrease the burden of oral conditions on the ER. The employment of oral health professionals in hospitals and emergency services might be of help in performing precise diagnoses and providing the most adequate therapy. Further, teledentistry should be powered, regardless of COVID-19, as it might help guide patients in all those situations in which a direct visit might be spared.

## Data Availability

The data presented in this study are available in Appendix A.

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
