# Peer review of "Dental Emergencies in an Italian Pediatric Hospital during the COVID-19 Pandemic"

_healthcare, 2022, doi:10.3390/healthcare10030537_

Round 1

Reviewer 1 Report

First of all I would like to congratulate the authors for this study. I would like to point out some things about the article:

In terms of article structure i would like to say:

The references should be in square brackets, and all the references should be in the same letter type as it is indicated on the template. 

There are no Conclusions on your article, you should write them. 

Also the contribution of the authors, if there is a ethic committee, how you could get the data from the hospital...

You should explain how your study could help in the hospital management, or in the ER management... If you just describe the ER numbers i think it is not enough.

I hope you can change this things

Author Response

Dear reviewer

thank you for your constructive feedback.

We worked on it

  1. The references should be in square brackets, and all the references should be in the same letter type as it is indicated on the template. Response 1: We corrected it
  2. There are no Conclusions on your article, you should write them. Response 2: we wrote down a new paragraph about conclusions that summarized our outcomes
  3. Also the contribution of the authors, if there is a ethic committee, how you could get the data from the hospital...Response 3: The study was approved by the ethical committee of Università degli Studi di Milano on December 14, 2021. We also described the contribution of each single author
  4. You should explain how your study could help in the hospital management, or in the ER management... If you just describe the ER numbers i think it is not enough. Response 4: we think that the employment of oral health professionals in hospitals and emergency services might be of help in performing precise diagnoses and providing the most adequate therapy. Further, teledentistry should be powered, regardless COVID-19, as it might help guide patients in all those situations in which a direct visit might be spared.

Reviewer 2 Report

Dear Authors, here are some suggestions regarding your work:

  • please, add obvious information in introduction regarding the presence of SARS-CoV-2 in the populity, such as where and when it started (Wuhan, Nov 2019), when it was first described in Italy (the country this paper originates from) and that it was found by WHO pandemic in 11.03.2020. This should be first 2-3 sentences within this work
  • Lines 38-39, please add information wheather they were closed due to Government restrictions
  • material and methods - please add information what kind of data did the Authors examine (pateint's paper records, computer records, computer statistics); please add information who (how many persons or add initials of them) and how searched the database
  • figure 1 - all months should be written staring with capital letter, the 2nd sentence in figure's title is not correctly stated and it is not understandable
  • please, divide tables and figures from the text, so they would be clearly seen in the text 
  • in table 1, please add a total number of visits (on the bottom) - 100%, although it should be considered weather this table should be used (if yes, maybe a good reason is to change the title to "hospital emergencies..."
  •  in the discussion session, there should be more information on the possible presence of oral cavity changes in SARS-CoV-2 infection, as it is widely described. The Authors do not refer too much on the oral mucosa changes, whereas they are widely discussed in disease itself and after the vaccination (refer to the references: Paradowska-Stolarz AM. Oral manifestations of COVID-19: Brief review. Dent Med Probl. 2021 Jan-Mar;58(1):123-126. doi: 10.17219/dmp/131989.; Orilisi, G.; Mascitti, M.; Togni, L.; Monterubbianesi, R.; Tosco, V.; Vitiello, F.; Santarelli, A.; Putignano, A.; Orsini, G. Oral Manifestations of COVID-19 in Hospitalized Patients: A Systematic Review. Int. J. Environ. Res. Public Health 202118, 12511. https://doi.org/10.3390/ijerph182312511; Mazur M, DuÅ›-Ilnicka I, JedliÅ„ski M, Ndokaj A, Janiszewska-Olszowska J, Ardan R, Radwan-Oczko M, Guerra F, Luzzi V, Vozza I, Marasca R, Ottolenghi L, Polimeni A. Facial and Oral Manifestations Following COVID-19 Vaccination: A Survey-Based Study and a First Perspective. Int J Environ Res Public Health. 2021 May 7;18(9):4965. doi: 10.3390/ijerph18094965.) - please try to discuss those to the fact, if the kids could be vaccinated in that time
  • line 166-170: please, try to think weather at that time there was the lockdown and that might have been a cause for smaller amount of injuries; Maybe it would also be valid to add the new term of teledentistry at that time (and therefore  reduction of the dental emergiences): Lewandowska M, Partyka M, Romanowska P, Saczuk K, Lukomska-Szymanska MM. Impact of the COVID-19 pandemic on the dental service: A narrative review. Dent Med Probl. 2021;58(4):539–544. doi:10.17219/dmp/137758
  • please define the term used in the paper: "painful caries" - do you mean any tooth decay that causes pain or pulpitis?
  • in the discussion session, there should be the information on the limitations of this study

Author Response

Dear Reviewer,

Thank you for your constructive feedback.

We worked on it.

  1. please, add obvious information in introduction regarding the presence of SARS-CoV-2 in the populity, such as where and when it started (Wuhan, Nov 2019), when it was first described in Italy (the country this paper originates from) and that it was found by WHO pandemic in 11.03.2020. This should be first 2-3 sentences within this work. Response 1: We added it in introduction
  2. Lines 38-39, please add information wheather they were closed due to Government restrictions Response 2 ok
  3. material and methods - please add information what kind of data did the Authors examine (pateint's paper records, computer records, computer statistics); please add information who (how many persons or add initials of them) and how searched the database Response 3: data were searched by initial diagnosis name (the results of the patient’s complain and triage) and discharge/admission final code and singularly checked in case of doubts. The following keywords were used: oral, oro-, facial, mouth, buccal, bucco- , dental, tooth, teeth, gingiva, gingival, stomatitis, aphthae, aphthous, abscess, caries, decay, lip, tongue, cheek.
  4. Figure 1 - all months should be written staring with capital letter, the 2nd sentence in figure's title is not correctly stated and it is not understandable Response 4: We converted the first figure in curve chart maybe clearer
  5. in table 1, please add a total number of visits (on the bottom) - 100%, although it should be considered weather this table should be used (if yes, maybe a good reason is to change the title to "hospital emergencies..." Response 5: We made it
  6. in the discussion session, there should be more information on the possible presence of oral cavity changes in SARS-CoV-2 infection, as it is widely described. The Authors do not refer too much on the oral mucosa changes, whereas they are widely discussed in disease itself and after the vaccination (refer to the references: Paradowska, Orilisi, Mazur ) Response 6: an additional paragraph focused on oral manifestation in SARS-CoV-2 infection has been written as you suggest. 
  7. line 166-170: please, try to think weather at that time there was the lockdown and that might have been a cause for smaller amount of injuries; Maybe it would also be valid to add the new term of teledentistry at that time (and therefore  reduction of the dental emergiences): Lewandowska M Response 7: yes, the decresed number of ER accesses was referable to lockdown period and governative restrictions. Teledentistry has been advocated as a useful tool for oral health education and promotion, remote diagnosis and monitoring. Ospedale Buzzi provided a telemedicine service for monitoring COVID-19 patients during the epidemic. Unfortunately it did not provide any teledentistry services.
  8. please define the term used in the paper: "painful caries" - do you mean any tooth decay that causes pain or pulpitis  Response 8:painful caries included any kind of pain from initial caries symptoms to pulpitis
  9. in the discussion session, there should be the information on the limitations of this study  Response 9: the present study includes some limitations. First, diagnoses were not made by oral health professionals and some misdiagnosing concerning oral diseases might be expected. Further, no data could be gathered concerning socio-economic characteristics, attitude towards oral health check-ups or eating habits of the sample population. It would have been interesting to know what kind of therapy followed the diagnoses of oral complaints, but such data are not available. Nevertheless, the same shortcomings applied before and during the pandemic.

We treasured your comments, hoping have been answered to all questions, you can find all the changes highlighted in yellow in the main text.

Thank you in advance

Marilisa Toma

Reviewer 3 Report

This is the review of the manuscript entitled "Dental Emergencies In An Italian Pediatric Hospital During The Covid-19 Pandemic".

In general:

  • I recommend using the template and also to read the information for authors again carefully. For example, there is no section "Data Analysis" as the main heading. https://www.mdpi.com/journal/healthcare/instructions

Abstract:

  • In my opinion, there is a conclusion missing within the abstract section.
  • Line 15: Duplication of the abbreviation explanation

Introduction:

  • Why are some references displayed in blue color?
  • Line 37/45: See also: Doi 10.3390/cancers13122892

Results:

  • From my point of view, it would make sense to adjust Figure 1. The bars of the single months should be better visible and comparable; this could be done by "shortening" the overall numbers („total"). Same for Figure 2. Besides, the numbers in Figure 2 are not displayed properly (and why now different colors for the years?). Btw, a curve chart would be appropriate to display the drop in admissions throughout the different moths.
  • In line 121, a space is missing and "table" ist written in lowercase. Please correct.
  • Line 162: The square bracket?
  • Please double-check the spelling of different synonyms for COVID/the virus. In line 165 SARS-CoV-2 is spelled wrong.
  • In my opinion, a real conclusion of this study is missing. Please add one.
  • What are the weaknesses/limitations of the study? Please add.

Author Response

Dear Reviewer,

Thank you for your constructive feedback.

We worked on it.

  1. I recommend using the template and also to read the information for authors again carefully. For example, there is no section "Data Analysis" as the main heading.  Response 1: We used the proper template and corrected it
  2. In my opinion, there is a conclusion missing within the abstract section  Response 2: Yes, we added it both in abstract and in main text
  3. Line 15: Duplication of the abbreviation explanation response 3: ok we corrected it
  4. Why are some references displayed in blue color? response 4:sorry maybe formatting error 
  5. Line 37/45: See also: Doi 10.3390/cancers13122892 response 5: we added it
  6. From my point of view, it would make sense to adjust Figure 1. The bars of the single months should be better visible and comparable; this could be done by "shortening" the overall numbers („total"). Same for Figure 2. Besides, the numbers in Figure 2 are not displayed properly (and why now different colors for the years?). Btw, a curve chart would be appropriate to display the drop in admissions throughout the different moths response 6: thank you,  we decided to convert the figure 1 in a curve chart maybe clearer
  7. In line 121, a space is missing and "table" ist written in lowercase. Please correct  response 7: ok
  8. Line 162: The square bracket response 8: ok
  9. Please double-check the spelling of different synonyms for COVID/the virus. In line 165 SARS-CoV-2 is spelled wrong.  response 9: ok
  10. In my opinion, a real conclusion of this study is missing. Please add one. response 10: Yes, we added it both in abstract and in main text
  11. What are the weaknesses/limitations of the study? Please add. response 11: 

    The present study includes some limitations. First, diagnoses were not made by oral health professionals and some misdiagnosing concerning oral diseases might be expected. Further, no data could be gathered concerning socio-economic characteristics, attitude towards oral health check-ups or eating habits of the sample population. It would have been interesting to know what kind of therapy followed the diagnoses of oral complaints, but such data are not available. Nevertheless, the same shortcomings applied before and during the pandemic.

We treasured your comments, hoping have been answered to all questions, you can find all the changes highlighted in the main text

thank you in advance

Kind regards

Marilisa Toma

Round 2

Reviewer 1 Report

Thank you for the improvements that you made on the article. I believe now is good to be published, just one thing, you should give at the end of the article the number for the project of the ethical cometee.

Thank You

Author Response

Dear Reviewer

thank you

-   you should give at the end of the article the number for the project of the ethical cometee.    Response 1   :  Attachment 7 ethical cometee december 14 2021 project number 131/21

Reviewer 2 Report

Thank you for all the corrections - the paper right now seem really nice, really. I would like to congratulate you on that topic! I think that rebuilding the discussion session and adding more information really improved the quality of that paper!

  • please, check line 213 (reference 95?) - while proof reading

Author Response

Dear Reviewer

thank you!

  • please, check line 213 (reference 95?) - while proof reading  response 1: I corrected it